# B and T Cell Epitopes of the Incursionary Foot-and-Mouth Disease Virus Serotype SAT2 for Vaccine Development

**DOI:** 10.3390/v15030797

**Published:** 2023-03-21

**Authors:** Qian Li, Ashenafi Kiros Wubshet, Yang Wang, Livio Heath, Jie Zhang

**Affiliations:** 1Key Laboratory of Veterinary Etiological Biology, National/WOAH Foot and Mouth Disease Reference Laboratory, Lanzhou Veterinary Research Institute, Chinese Academy of Agricultural Sciences, Lanzhou 730046, China; 2Department of Veterinary Basics and Diagnostic Sciences, College of Veterinary Science, Mekelle University, Mekelle 2084, Tigray, Ethiopia; 3Transboundary Animal Diseases: Vaccine Production Programme, Onderstepoort Veterinary Research Institute, Agricultural Research Council, Pretoria 0110, South Africa; 4Hebei Key Laboratory of Preventive Veterinary Medicine, College of Animal Science and Technology, Hebei Normal University of Science and Technology, Qinhuangdao 066004, China

**Keywords:** foot-and-mouth disease virus, serotype SAT2, vaccine, epitopes, immunoinformatics

## Abstract

Failure of cross-protection among interserotypes and intratypes of foot-and-mouth disease virus (FMDV) is a big threat to endemic countries and their prevention and control strategies. However, insights into practices relating to the development of a multi-epitope vaccine appear as a best alternative approach to alleviate the cross-protection-associated problems. In order to facilitate the development of such a vaccine design approach, identification and prediction of the antigenic B and T cell epitopes along with determining the level of immunogenicity are essential bioinformatics steps. These steps are well applied in Eurasian serotypes, but very rare in South African Territories (SAT) Types, particularly in serotype SAT2. For this reason, the available scattered immunogenic information on SAT2 epitopes needs to be organized and clearly understood. Therefore, in this review, we compiled relevant bioinformatic reports about B and T cell epitopes of the incursionary SAT2 FMDV and the promising experimental demonstrations of such designed and developed vaccines against this serotype.

## 1. Introduction

Foot-and-mouth disease (FMD), alternatively called aphthous fever, is an easily transmitted disease that infects ungulates such as cattle, sheep, and pigs. It is drastically harmful to the livestock industry. Currently, it affects more than 70 susceptible animal species in 80 countries [1]. Interestingly, on one hand, no country is far away from the threat due to the highly contagious nature of the virus and the intensified international trade of animals and animal products [2]. For this reason, the World Organization for Animal Health (WOAH) calls for international solidarity along with independent regional programs to effectively control the disease [3]. On the other hand, FMD has been controlled successfully in many countries by implementing comprehensive strategies, such as obligatory vaccination of susceptible animals, control of animal movement, and slaughtering of infected animals [4]. Obviously, it is inevitable for virologists and field veterinarians to make claims about the unsatisfactory vaccination success stories since the start of FMD vaccine development 70 years ago [5]. This lack of success may be due to the persistent genetic evolution of the virus. 

A unique property of the aphthous virus among the Picornaviridae is the presence of multiple serotypes (O, A, C, Asia1, South African Territories (SAT) 1, 2, 3) unluckily deficient in cross-immunoprotection [6]. Moreover, there are multiple subtypes, lineages, and topotypes (Table 1) due to the high mutational rate of the virus, continuous antigenic drift during the process of replication [6], and quasi-species dynamics [7]. The hypervariable regions on surface-exposed loop areas of the virus structure [8], no matter how high and how less, create genetic and antigenic divergence among FMDV serotypes [9]. 

In addition, the South African types (SATs) have shown higher interserotype sequence diversity than Eurasian types [10]. Furthermore, nucleotide and amino acid level intratypic variations are also more common in SAT types than in the European origin (serotype O, A, and C) [9,11]. FMDV’s persistent genetic evolution, potentially higher genetic and antigenic variation, and serotype instability are among the challenges for virologist and field veterinarian success with different types of vaccination designs and vaccines in use since the 1930s [5].

Thus, regular monitoring of the circulating topotypes and lineages of FMDV serotypes in endemic regions is very crucial for matching the field and vaccine strains [10]. This would help to implement a selection of appropriate vaccine strains for strategic control and prevention practices in endemic regions.

Generally, the genome of a mature virion particle with a diameter of about 30 nm is about 8.4 kilobases, which contains a long open reading frame (ORF), with no cap structure at the 5′ terminal, but embedded with a poly A tail at the 3′ terminal [1]. Each virion contains only a copy of the genome and protein shell composed of 60 copies of four structural proteins, namely VP1 (1D), VP2 (1B), VP3 (1C), and VP4 (1A) (Figure 1). Thus, the structural proteins play an important role in determining the potency of the vaccines.

Epidemiological challenges to control FMD vary in developing and developed countries. Regardless of the unclear natural endemic cycle of FMDV, virus reservoirs between outbreaks, and the undetermined role of convalescent carrier animals in disease epidemiology, the most important challenges in developing countries are related to the size of susceptible populations, along with dynamicity of the market chain and uncontrolled animal movement [19]. Besides overcoming a number of challenges, control strategies based on sustained vaccination have proven to be an efficient control method. However, lack of potential cross-protection, the risk of reversion, and poor reactions to vaccine components or inconsistency between vaccine batches remain as unavoidable potential drawbacks of Jenner’s first-generation viral vaccine [20,21]. Furthermore, traditional FMDV vaccines do not provide enough protection for vaccinated animals, require re-vaccination every 4–12 months due to limited shelf life, need the growth of a virulent virus, posing a threat of escape from manufacturing sites, and interfere with the non-structural protein (NSP)-based serological differentiation of infected from vaccinated animals (DIVA) [17,22].

### Brief Evolutionary History of FMDV Vaccines (1930–2022)

FMDV vaccines have evolved considerably over time. Conceptually, Jenner’s type of FMD vaccines were among the first animal vaccines developed in the 19th century [23], and notions of mass vaccination policies in the 1950s enlightened countries in, for example, North America, Western Europe, and parts of Asia, leading them to eradicate this feared disease [5], and the commercialization of inactivated vaccines was inaugurated [24]. A year after the founding of the World Organization for Animal Health (WOAH) as Office International des Epizooties, the first scientific experimental vaccine against FMD was made, in 1930, by Waldmann and his co-workers [25]. Until the end of the 1940s, the formaldehyde inactivation technique of virus-infected organs was used as the best approach for immunization of cattle [26]. In 1950, this inactivation method, using in vitro culture technology on viable tongue epithelium, allowed FMD virus growth for vaccine production on a large scale [27]. Neither mass vaccine production nor a systematic annual vaccination program against FMDV were begun till early 1950 [28]. However, in 1953, the Netherlands launched a yearly cattle vaccination program using the ‘’Frenkel type’’ of vaccine. BHK 21 cells were a breakthrough, thanks to Macpherson and Stoker, and, in 1962, scientists [29] started to grow FMDV in the BHK 21 monolayer, with the objective of producing vaccines on a large scale. Afterwards, the 1970s’ binary ethyleneimine (BEI) inactivation method launched, and immunization was enhanced by oil adjuvants [30]. About 50 to 60 years of preliminary vaccine production trials by different scientists around the globe served as a baseline for the current novel vaccines, which do not use inactivated antigens. However, after a total of 90 years of research, there is no effective vaccine that produces sterile and solid immunity for FMD [31]. Like any other human virus vaccines, based on the course of evolution and specific characteristics, spectrum of coverage FMD vaccines are divided in to three generations (Table 2).

The advancement of knowledge in the field of bioinformatics seems to shift both medical and veterinary traditional vaccine development approaches into a modern vaccine design in order to enhance the protection and avoid risks. The bioinformatic tools could provide alternative rational choices to isolate the ideal structural protein components desired for production of the humoral and cellular immune response. In addition, such vaccine designs have a potential advantage in their ability to focus immune responses on discrete epitopes, increasing safety, potency, and breadth of multiple serotypes [57,58,59]. SAT serotypes of FMDV have higher genetic diversification, so that they demand such robust vaccine research efforts. However, the majority of the investigations on epitope identification and epitope-based vaccine engineering were conducted on Eurasian serotypes of FMDV. Only a few discrete reports and information on SAT serotypes’ B and T cell epitopes and experimental demonstrations of the designed vaccines are available. This may be due to technical and practical issues along with economic resource issues in many countries where this serotype is endemic, or there may be less focus given by researchers and research institutes from SAT FMDV-free countries.

Nevertheless, a clear understanding of concepts related to available B and T cell epitopes and molecular engineering practices is important for the development of novel and powerful epitiope-based vaccines that could protect animals from the highly incursionary SAT2 FMDV. In this review, therefore, we collected information on new and applicable concepts and reports related to B and T cell epitopes of incursionary SAT2 FMDV and their promising impact on vaccine design and development, and available evidence on protection potential. Epitope identification methods and epitope-based vaccine designing strategies against Eurasian serotypes of FMDV are also briefly discussed.

## 2. Molecular Epidemiological Challenges of FMDV Serotype SAT2

The evolutionary mechanisms of RNA viruses (recombination, positive, and negative selection, and random genetic drift constraints) are also considered to be the cause of the rapid evolution of FMDV in East Africa. More specifically, interspecies transmission of FMDV contributes to the rapid evolutionary divergence of the virus, mainly because of the variation of the host-specific virus genetic coping mechanism [60]; for instance, different ecologies of host species (differential selection pressures between host species), a lack of infectivity of the virus in new hosts (nucleotide-divergence of carrier animals and virus during clinical phase varied from 0.1% to 1.3%) [61], or interference by host immunity. Evolutionary dynamics have arisen from deletions within the coding and non-coding regions of the FMDV genome and recombination involving exchange of the capsid-coding region between serotypes and intratypes [62].

For the last many years, SAT FMDV had not been spreading across the equatorial region, as the virus probably required a special environment in Africa. However, in recent years, many reports indicate that SAT types, in particular SAT2, have been detected in the geographical settings of Libya, Egypt, the Palestinian Autonomous Territories (PAT), and Bahrain. As a result, serotype SAT2 is considered a potential threat to the development of the livestock industry in Northern Africa and neighboring countries in the Middle East [63]. As well, an outbreak of SAT2 FMDV would possibly pose a serious risk to the Europe–Asia boundary and beyond, where intensive pig farming systems are most popular. A study showed that experimentally inoculated pigs can become infected with an SAT2 serotype [64]. However, no single report has yet come from these areas. Therefore, the vaccine research in SAT2 FMDV-free areas is mostly related to O, A, C, and Asia1. This might be one cause for the absence of alternative multiple efficient diagnostic technologies and safer vaccines against SAT FMDV in the world [65].

WOAH’s FMDV outbreak study reports from 2000 to 2010 showed that the serotype SAT2 virus dominated outbreaks, together with types O and A [66]. This serotype was also recorded as the second-most-dominant circulating type in a 10-year (2008–2018) FMD outbreak study report from Ethiopia [8]. Another study by Diab et al. in Egypt (2013–2014) ranked SAT2 as the second-most-abundant serotype next to serotype O [67]. Likewise, outbreak reports from the Ministry of Agriculture of Egypt from 2012 stated that 4658 animals, mostly calves, out of the 80,000 cattle and buffalo, died from SAT2 FMDV suspected cases [68]. Most of these studies suggested that animal markets played a critical role in the spread and distribution of the virus [19].

In 1990, 2000, 2003 and 2012, SAT2 FMDV incursionary outbreaks surged in Northern Yemen, Saudi Arabia, Kuwait, Libya, Egypt, the Palestinian Territories, Libya, and Bahrain, respectively [19,69]. The isolates from the Egyptian and Palestinian region had a close genetic relationship to a similar type of virus in Eritrea in 1998, leading to speculation about the source of the new jumps of SAT2 virus to the northern African and Middle East region [70]. The uncommon epidemiological jumps of this virus from Sub-Saharan Africa to SAT FMD-free boundaries may be farsighted as a direct **or** potential threat to the pig industry in Eurasian countries such as China, because animal experiments have proven that pigs could largely be susceptible hosts for the serotype SAT2 virus [64,71]. Therefore, there is a risk that this kind of cross-border introduction (Figure 2) of this serotype may possibly happen in China due to the increasingly frequent trade transactions between China and African countries.

In addition to the overall existing epidemiological challenges of FMDV, interspecies transmission of SAT serotypes is present between cattle–antelope and buffalo–cattle, and other species, where the buffalo remains the ultimate source of infection for susceptible cloven-hoofed animals [72,73]. The circulation of SAT1 and SAT2 viruses in buffalo continues to produce mutations, which consequently leads to viral antigenic variation in Southern African regions [74]. Hence, the higher genetic diversity in SAT serotypes may be the basic molecular reason for having many genetic and antigenic variants as blueprints for different geographic origins, leading to the failure of vaccines in cross-protection.

The serotype SAT2 is composed of 14 topotypes [74], designated from I to XIV, with 80% sequence identity in the VP1 coding nucleotide sequence [75,76]. This topological genetic diversity is possibly because of higher intratypic recombination [77]. The VP1, which carries critical epitopes for induction of an immune response, plays a dominant role in genotyping. For instance, the Egyptian SAT2 topotype VII was further subtyped into two different subgroups (ALX and GHB) based on the nature of VP1 amino acid sequence diversity [74]. Due to the high natural mutation rate of this FMDV type, the possibility of the strain emerging with new topotypes and lineages is very high.

F.N. Mwiine and his co-researchers reported a phylogenetic analysis of the SAT1 virus isolated from toptype I and IV and the SAT2 virus from toptype VII, IV, and X from six and four regions, respectively. The results indicated a possible interspecific transmission at the wild animal–livestock interface. This 3-year study provided knowledge of the geographical distribution of the serotypes of foot-and-mouth disease virus isolated in Uganda, where different SAT2 topologies were identified at the same time, and also reflected the epidemiological complexity of SAT2 FMDV [78].

The multiplicity of the topotypes of SAT2 FMDV could therefore complicate control attempts. Thus, the immunity induced by vaccination against the lineage of one FMDV topotype may fail to protect against infection with viruses of the other topotypes in different geographical areas. Thus, continuous monitoring on the circulating FMDV topotypes and lineages is crucial to ensure the application of effective vaccines and other appropriate control measures [79]. Moreover, understanding the molecular epidemiology and immunoinformatic features of the emerging strains could also help to develop safe, stable, and protective vaccines against multitopotypic SAT2 FMDV.

## 3. Epitope-Based FMDV Vaccine Development Approaches

Development of broadly reactive vaccines which can confer immunity against multiple FMDV serotypes remains a difficult job. Similarly, DIVA vaccine issues, inefficient protection of the inactivated FMDV vaccines, thermal instability of vaccine components at field level, and a short period of protection are also considerable problems [80,81,82,83]. These and other factors greatly altered the perspectives of experts on finding more effective and safer second- and third-generation vaccines against FMDV [84]. As an alternative, epitope-based vaccines could complement or even substitute for a number of commercially available classical vaccines that have failed to protect against FMD outbreaks, as has happened in Saudi Arabia [85] and other endemic regions.

Epitope-based subunit vaccines are mentioned as one of the third-generation vaccine approaches, where the predicted epitopes could elicit high antibody (Ab) titers, and increase activation of CD4, CD8, and natural killer (NK) cells [86,87]. These types of vaccines are produced by epitopes using different design and construction methods. One of the fast-growing areas of knowledge on epitopic vaccine design is the grafting or recombining of anticipated antigenic epitopes onto viral DNA backbone [88]. Epitope-grafting approaches include the development of a multi-epitope-based subunit vaccine (MESV) that could fully activate both cellular and humoral immune response against multiple serotypes [87,89,90]. This can deliver a synergistic protection effect against several serotypes or subtypes of FMDV [50]. Single or multiple epitopes can be displayed in a highly ordered and repetitive array on nanoparticles and virus-like particles (VLPs) to elicit potent immune responses [20,91,92,93]. Since the antigenic sites in different FMDV serotypes are structurally and functionally the same, and only vary in positions and molecular constitution [20,94], an MESV approach is ideally suited for all FMD serotypes, and for the multitopotypic serotype SAT2 in particular [21,95]. Many experimental trials have so far been conducted to synthesize chimeric peptide vaccines against FMDV by incorporating neutralizing B and T cell epitopes [96]. Some of them were shown to confer protection against FMDV challenges [97,98]. However, up to now, only a few peptide vaccines are currently licensed and commercially available in the world [99]. Moreover, most of these trials were conducted on serotypes O, A and Asia1.

Epitope prediction is the primary job of the development plan of epitopic vaccines. Epitope prediction methods from given viral antigens can vary for different practical reasons and intended purposes. Diverse and robust algorithm-based tools have now simplified the protein structure prediction and antigen–immunoreceptor interactions. In addition, these tools ultimately help in designing and evaluating the epitopic vaccines [56]. To date, there are about 33 T cell epitope and 22 B cell epitope prediction tools accessible online for free public use. T cell epitope prediction methods are more advanced and reliable than B cell prediction tools [100]. Mapping of epitopes from a given structural protein can also be performed by using microarrays and enzyme-linked immune absorbent spot (ELISPOT) or enzyme-linked immunosorbent assay (ELISA) techniques [8]. Mutagenesis is another possible high-throughput epitope scheming approach to rapidly improve predicting conformational epitopes on structurally complex proteins [101]. Except for the question of accuracy, the in silico method is the most cost-effective mode of identifying B cell antigenic determinants in a target vaccine candidate [102]. In general, development of epitope-based vaccines against RNA viruses like FMDV is not easy due to the challenge to selecting epitope peptides from widely varying serotypes and diversified host cellular immune systems. Nonetheless, the advancement in immunogenetic engineering and manipulations of viral immunodominance regions would ultimately help to optimize and rationally design an MESV, as detailed below (Figure 3) [103]. Hence, the mutagenesis approach and other tools mentioned in the above could apply to predict conserved and neutralizing epitopes of FMDV.

The majority of the FMDV epitope prediction activities have been implemented on serotypes A, O and Asia1 by exploring the VP1 structural protein as illustrated in (Table 3). For instance, Momtaz et al. [104] predicted a total of 11 B and T cell epitopes of serotypes A and O by a combination of the evolutionary and computational approach, based on outbreak isolates in India and Bangladesh, respectively. Yun et al. [105] also predicted secondary structures, such as α-helix, β-sheet, corner, and random curl, based on the sequence of the VP1 protein of the FMDV O LZ02 strain using Garnier–Robson [106] and Chou–Fasman methods [107]. In both reports, analyses of the surface probability plot and antigenic index of the VP1 protein, such as hydrophilicity, flexibility, accessibility, and antigenicity, were executed using Kyte–Doolittle [108], Karplus–Schulz [109], Emini [110], and Jameson–Wolf [111] methods, respectively. In addition, Yun et al. [105] reported that dominant B cell epitopes were found in the VP1 protein. Aggarwal et al. [112] also identified five antigenic epitope sites of serotype O. Of course, amino acids (aa) 140–160 and 200–213 on N- and C- termini of VP1 are the most immunogenic regions [113].

Type A’s immunogenic epitopes contain two major sites, aa 140–160 of VP1 and xxx in VP3 residues, and another two minor sites located at the C-terminus of VP1 and aa 169 near VP1 [122]. Likewise, type A has two neutralizing epitope sites, located at the C-terminus of VP2 and VP1 [119]. In general, the FMDV epitopes most commonly reported are within a highly variable G-H loop in VP1 [120,123,124]. Borley et al. also predicted epitope sites of serotypes A and C at aa 196 over the EH-EI loop [125]. The B cell epitopes of three structural proteins (VP1, VP3, and VP4) of type Asia1 were predicted by Zhang et al. [126] with a similar approach. Another epitopic region of serotype C was found at positions aa 163–176 of VP2 [123] and aa 127–140 (conformational epitope) of VP3 [127]. Likewise, experiments showed that the fusion proteins consisting of B and T cell epitopes of serotype O could stimulate the humoral and cellular immune response in guinea pigs [128,129]. Luis et al. suggested that the TrpE fusion proteins containing portions of the C-terminal region of VP1 of type O activated a neutralizing antibody response and conferred full protection in guinea pigs challenged with homologous virus [130]. However, fusion proteins of tandem repeat epitopes corresponding to amino acid positions VP1 (133–158) and VP4 (17–31) of type Asia1 failed to offer complete protection in guinea pigs [131]. On the other hand, partial protection with linear peptides containing B and T cell epitopes of serotype A was recently reported for cattle [132].

## 4. B and T Cell Epitopes of SAT2 FMDV and Vaccine Design and Developments

Given the potential anticipated impact on global animal health, many efforts are essentially required to be expended on the multiple variants of serotype SAT2 to identify and elucidate their conserved immunogenic sites for multi-epitope-based or epitope-driven vaccine design and development [109]. Furthermore, low antigenic coverage of the present inactivated vaccine against SAT2 field strains demands serious measures on the development of a multi-topotype-specific SAT2 vaccine that will provide protection against a wide range of antigenic types in the field [8]. This is a great implication of SAT2 epitopes in vaccine design and development if the epitopes are going to be predicted periodically. In addition, an MESV is believed to be able to substitute for the classical inactivated vaccine currently in use for the prevention and control of FMD in the future [133].

VP1, VP2, and VP3 are antigenic domains of FMDV and are highly prone to antigenic variation [123]. Basically, the variation in the VP1 region among FMDV serotypes impaired the development of vaccines that could offer cross-protection between different serotypes [134]. Unlike VP1, with only about 80% identity, an old report from F. Brown [135] on capsid protein sequencing data shows that VP4 is a highly conserved molecule (98%), and VP2 and VP3 share 90% homology. It is widely reported that VP1 plays an important role in virus attachment and entry, protective immunity, and serotype specificity. Moreover, it is the most studied structural protein in SAT2 and other serotypes. In fact, VP1 ranges in size from 217 to 221 aa length, which contains sites aa 140–150 and aa 166–170 for insertion/deletion in different strains of FMDV. It was recognized that only 26% of the nucleic acids in VP1 coding sequences are stable [62]. A study report from Vosloo et al. [74] indicated that the nucleotide replacement rate of SAT2 FMDV on the VP1 coding region was 1.64% every year. Emad et al. [67] also reported that the mutation sites of the SAT2 FMDV strain obtained in Egypt were mainly concentrated in amino acid sites 131–149, 156–166, and 206–212. Moreover, aa 159 located in the immunogenic region at the surface-exposed VP1 protein mutated from His to Arg [136]. Hence, the main reason why peptide vaccines produced a very weak immunological efficiency is possibly due to antigenic heterogeneity generated in the VP1 protein across subtypes. DiscoTope, Ellipro, Epitopia, and SEPPA are the most commonly utilized epitope prediction programs in identification of novel epitopes in the structural proteins of SAT FMDV [125,136]. Further, different methods were applied for mapping the epitopes of SAT2 FMDV, as indicated in (Table 4).

Accordingly, a novel epitopic sequence was predicted by Borley and his co-workers [125] at position aa 95–105 on the EE-EF loop of VP1 for serotype SAT1 and SAT2. The B cell epitope, located at aa 58–71 of VP3, reported as an antigenic site of serotype O, was also believed to play a key role in the formation of SAT2 antigenicity [62]. Other T cell epitopic regions on VP1 at position aa 199–211 and at C-terminal aa 200–213 were considered as essential antigenic sites of serotypes O and A. Those regions may also be important epitopes in serotype SAT2. The G-H loop (aa 140–160) is universally characterized as a primary B epitope to all FMDV serotypes [95,124,138]. Similarly, Liu et al. confirmed that aa 89–105 in the VP2 protein is a conserved discontinuous epitope among all seven serotypes, according to the analyzed sequence alignments [139]. On the other hand, a unique epitope site at aa 66–71 on the EB-EC loop of VP3 was identified across all SAT serotypes. An equivalent loop was found at aa 60–70 for serotypes A, O, and SAT1 [125]. In addition, Borley et al. [125] identified a structural loop named EH-EI loop of VP3 for serotypes SAT1 and SAT3, where a novel epitope at aa 193–197 was encountered. Wu et al. [140] analyzed the hydrophilicity and flexibility of each amino acid on three structural proteins, VP1 to VP3, to determine the possible epitopes of serotype SAT2. During epitope prediction, the flexibility and hydrophilicity clarify the binding nature of antigenic sites to the antibody [92,141]. For example, a flexible loop (like the G-H loop) is a secondary structure that normally helps with the integrity of capsid proteins, which, in turn, play a significant role in sequence diversity among serotypes [121]. Wu et al. [140] reported that the 14 aa length polypeptide chain located at positions aa 163–176 on VP2 and aa 127–140 on VP3 of SAT2 FMDV showed a strong immunogenicity, and they also added that aa 132–146 of VP1 showed a good immunogenicity reaction. Malirat et al. [127] found a conformational epitope at aa 127–140 on VP3. Residue aa 79 of VP2 was also reported as an antigenic site in SAT2 FMDV, although the details in the immunity are unknown [21]. Additionally, aa 144–154 and aa 200–213 of the GH loop and C-terminus of VP1, respectively, were also identified as conformational epitopes of SAT2 FMDV [137]. Carrillo et al. [62] performed a genomic comparison of FMDV, pointing out that VP4 is the most conserved protein, of which 81% of aa is invariant, including N-terminal myristylation sites. Moreover, Carrillo et al. [62] also reported a T cell epitope on the VP2 protein at residue sites aa 60–73 and aa 70–73 of SAT2 and O FMDV, respectively. A study by Maree et al. [8] predicted two specific amino acids in the VP4 protein of SAT2 and SAT1 viruses as possible neutralizing epitopes. Similarly, T cell epitopes at aa 20–35 of VP4 were identified for the SAT2 virus.

Opperman et al. [136] applied chicken single-chain antibody fragments (scFv) to map antigenic determinants on an SAT2 FMDV. Position aa 159 of the VP1 protein located at the C-terminal base of the G-H loop of the SAT2/ZIM/7/83 virus is very important for neutralization. Two years later, Opperman et al. [137] identified aa 71–72 and 133–134 of VP2 and aa 48–50, 84–86, 109–111, 137–140, 157–160, 169–171, and 199–201 of VP1 as putative epitopes for SAT2 FMDV. SAT serotypes shared more predicted epitopes with known epitopes on type A virus than any of the other serotypes [130]. On the other hand, some predicted putative epitopes, such as aa 102 of VP1, aa 87–90 of VP2, aa 58–59 of VP3, and aa 67–70 of VP3, are considered as consensus epitopes among all serotypes. Furthermore, Michelle et al. [136] predicted 11 epitopes shared among sequences taken from 15 topotypes of SAT2 FMDV. Following the multiple sequence alignments and homology modeling of the sequences, consensus epitopes among all SAT serotypes were found. These could be used as potential targets for the rational design of cross-reactive MESV. To summarize, a list of known epitopes on serotype SAT2 are described in Table 5 below for future research aiming tow the development of epitope-based vaccines.

An increase in antibody production after activation of B cells is typically associated with CD4+ T cell-mediated host response. Synergy between B and T cell epitopes plays an important role in inducing effective immune response.

Experiments showed that a T cell epitope located in aa 21–35 of the NSP 3A, which is highly conserved among different FMDV serotypes and possesses the capacity to induce T helper activity, could allow cooperative induction of anti-FMDV antibodies by B cells [144]. Synergy between a universal FMDV T cell epitope and B cell epitope in VP1 ensured a higher degree of serotype-specific activity by in vitro stimulation and increased host protection against the infectious particle [138,145]. Hence, Min et al. [146] also advised enclosure of the immunopotent T cell epitope of 3A in epitope-based FMDV vaccine design and development to promote a CD4^+^ T cell response and stimulate cytokine secretion, such as interferon (IFN), tumor necrosis factor (TNF), and interleukin (IL)-17A. This strategy could also be applied to prepare an SAT2 vaccine because the amino acid sequences of 3A exhibit limited antigenic variability across all FMDV serotypes. Furthermore, animal experiment reports since the late 20th century realized that the recombinant proteins of immunogenic epitopes located in residues aa 141–160 and aa 200–213 of the VP1 protein could confer full protection against a virus in laboratory animals, but limited protection against large animals’ viral challenge [147,148]. However, it was reported that MESV against type O could provide full protection in swine after challenge [47,133,149,150,151], and commercial epitope vaccines protecting swine from type O infection have been applied in China [99]. Recently, Chang et al. [88] inserted conserved neutralizing epitope 8E8 of serotype O into bovine parvovirus (BPV) capsid protein VP2, which could be self-assembled into VLP, and generated ten chimeric VLPs with similar shape and size to wild-type BPV VLP. However, only two chimeric VLPs, rBPV-VLP-8E8 (391) and rBPV-VLP-8E8(395), were able to induce neutralizing antibodies. Among the two hybrid VLPs, the 8E8 epitope was inserted into positions aa 391–392 and 395–396, respectively, in the virable regions of BPV VP2. Thus, insertion positions in the carrying vectors, such as BPV VP2, also have significant impact on host immune responses besides the B and T cell epitopes. Those strategies for MESV against FMDV should be taken into account when the epitope-based vaccines are designed for serotype SAT2.

### Adjuvants and Delivery System

Since the 1930s, a variety of adjuvants have been discovered to enhance immune responses to FMDV vaccines, particularly for the new generation vaccines (which are poorly immunogenic) [152], and [153] to strengthen their immunogenicity and extend the duration of protection. In earlier times, immunologists did not have a clear understanding of the mechanism of adjuvants in vaccinology, and they used to call it a ‘dirty little secret’ [154]. Generally, in today’s vaccines (second- and third-generation vaccines) against FMD, adjuvants are designed to elicit strong inflammatory signals targeted at APCs, linking the two arms of the immune system (Figure 4). Shortly, understanding of the combination PRR ligands and factors able to activate DCs became significantly important to enrolling selective adjuvants in immunization [84]. Adjuvants and delivery systems shared a common feature in maximization of antigen concentration in immune organs to maximize the job of APCs. More recently, particulate delivery vehicles (nanoparticle polymers (NPs), such us Mesoporous silica nanoparticles (MSNs) [155], chitosan nanoparticles (CS) [68], gold nanoparticles, and L-Lactide-co-glycolic acid (PLGA) [156,157], have been used in inactivated and virus-like particle FMD vaccines, and the results indicate that these innovative delivery systems were effective in different degrees. Details of the different types of adjuvants are described in Table 6 below.

## 5. Conclusions

Multiple topotypic SAT2 FMDV is highly prone to antigenic variation and has been periodically appearing with new variants. This eventually led to the low cross-protection (intratypic protection) and inefficiency of the current classical vaccines. The epitope-based vaccination approach has been promisingly conferring immunity against all intratypes, and possibly all serotypes, to overcome the drawbacks mentioned above. Thus, appropriate designing and assessing of epitope-based vaccine candidates in generating effective immune response is very critical to their success in successive clinical trials. To work with this approach, knowledge of B and T cell epitopes is vital. At present, most research on epitope-based vaccines for FMDV are related to Eurasian serotypes, but there is little comprehensive information about serotype SAT2. In this review, we collected data and reports on B and T cell epitopes of the incursionary SAT2 FMDV and their implication for vaccine design and development. The pieces of information about potential immunogenic epitopes of SAT2 FMDV in our review can offer a clear foundation for ongoing efforts toward vaccines against serotype SAT2.

## Figures and Tables

**Figure 1 viruses-15-00797-f001:**
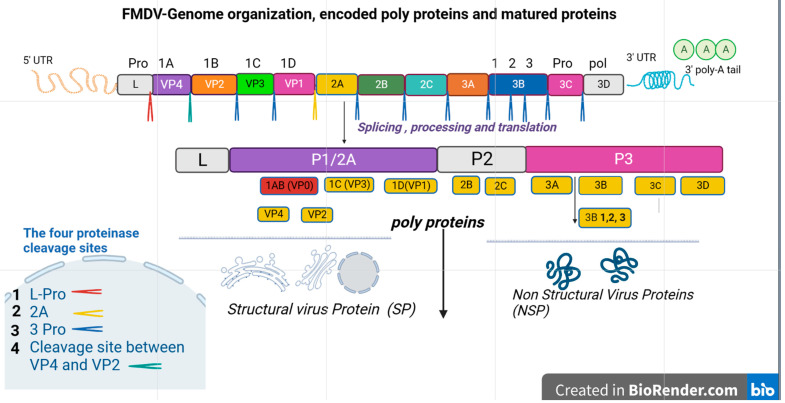
This simple schematic diagram shows the genome organization of FMDV, the encoded polyprotein, and matured proteins. It describes the long process of protein cleavage. The single ORF is illustrated in the box, and the viral proteins are named according to Rueckert and Wimmer’s nomenclature of picornavirus proteins [18]. Inside the boxes, there is a leader protein (L pro), four structural viral proteins (VP1-4), and seven non-structural proteins (2A-C, 3A-D). In addition, cleaved protein products and the cascade of cleavage are also shown in the diagram. The mature functional protein elements after cleavage are categorized as structural and non-structural proteins. The main cleavage sites are also shown in the box at the left side. The right and left untranslated regions (UTR) of the open reading frame are 3’ UTR and 5’ UTR respectively. The right extreme flank is shorter than the left one.

**Figure 2 viruses-15-00797-f002:**
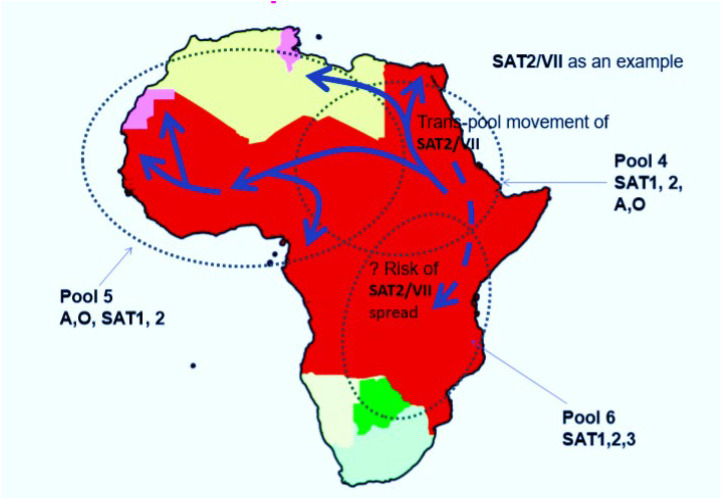
Incursionary features of FMDV serotype SAT2. In this map, SAT2 topotype VII was taken as an example. The directional arrows on the map show the epidemiological dynamics of SAT2 across different African regions, jumping the WOAH demarcated pools.

**Figure 3 viruses-15-00797-f003:**
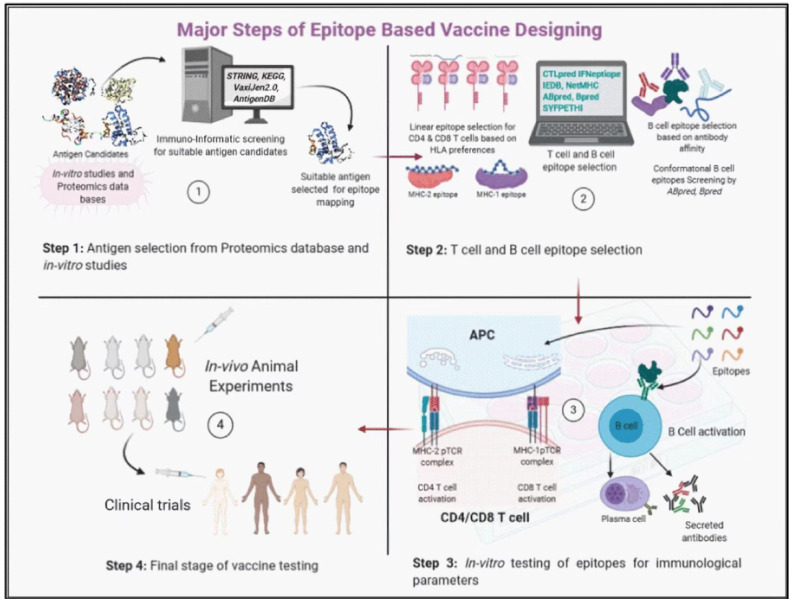
The illustration describes successive steps in epitope-based vaccine designing (this figure was adopted from Aryandra Arya and Sunil K Arora [103]. (1) In vitro antigenic epitope selection, identification, and analysis by using multiple immunoinformatic software from proteomic databases; (2) immunogenic B and T cell epitope selection by epitope insertion and sequence analysis; (3) synthesis of the designed neutralizing epitopes in the form of particulate, and evaluation of humoral and cellular immune responses by in vitro testing; (4) In vivo animal immunization to analyze the antibody responses to functionally characterize the anticipated neutralizing Abs- and cell-mediated immune responses.

**Figure 4 viruses-15-00797-f004:**
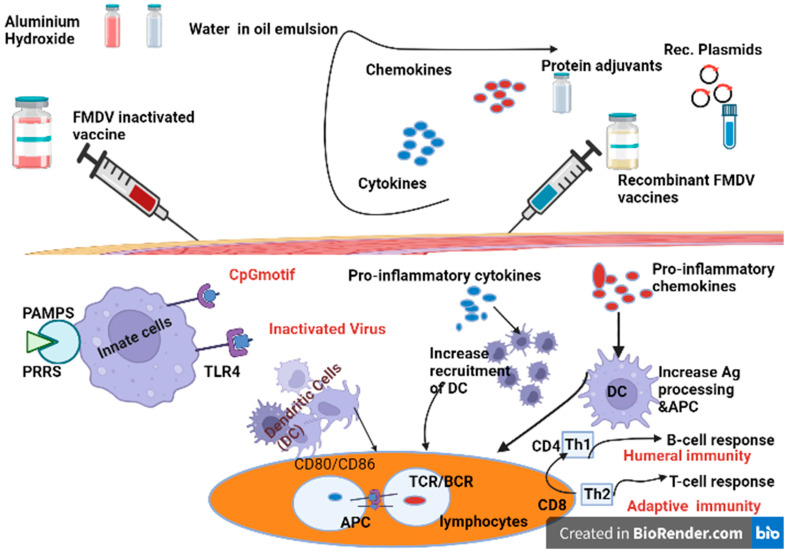
Adjuvants targeting both innate and adaptive immunity and the mechanism action. (**A**) Innate immunity; molecular adjuvants (cytokines, chemokines, and dendritic stimulating molecules) incorporated as a gene in the same plasmid and, when co-delivered with the vaccine, can elicit efficient DC recruitment, activation, and Ag presentation to T cells. The CD8+ T lymphocytes are authorized by dendritic cells to become effector CTLs. For this, antigens need to be taken up, processed, and presented by dendritic cells (DCs) in association with MHC molecules. Cytokine adjuvants help sustain CD25+ regulatory T cells (T-regs) of the CD4+ cells. This can significantly improve the immune response. (**B**) Vaccine with adjuvants, such as oil-in-water emulsions and almunium hydroxide, can stimulate the adaptive arm of the immune system. The maturation of DCs are firstly recognized by CD4+ and CD8+ T cells. CD4+ T lymphocytes undergo a clonal expansion into two distinct T-helper subpopulations following contact with the MHC I-peptide complex. TH1 and TH2 cells stimulate the humoral and cell-mediated immune response through a different cytokine expression pattern. TLR4 agonists are considered to be the main ligand to activate DCs. This kind of adjuvant elicits a TH2-type functional immune response with preferential IgG1 antibody production [157,158,159]. CTL: cytotoxic T lymphocyte; DC: dendritic cell; PAMP: pathogen-associated molecular pattern; PRR: pattern recognition receptor; TLR: toll-like receptor (TLR is a receptor family belonging to PRRs); Treg: regulatory T cell; Th: T helper cell.

**Table 1 viruses-15-00797-t001:** Summary of the serotypes, subtypes, lineages, and topotypes of FMDV.

Serotypes	No. Topotypes	Topotypic Nomenclature	Subtypes	Lineage	Description of Geographic Location and Genetic Diversity
O	11	I Cathay, II Middle East–South Asia (ME–SA), III South East Asia (SEA), IV Europe-South America (Euro–SA), V Indonesia-1 (ISA-1), VI Indonesia-2 (ISA-2), VII East Africa (EA1), VIII East Africa (EA2), IX East Africa (EA3), X East Africa (EA4) and XI West Africa (WA) [12,13].	-		Except for ISA-1 and ISA-2, which is found only in Indonesia, almost all of the other topotypes have been reported in different parts of the world [14].
A	3	Africa, Asia, and Europe-South America (Euro–SA) [12].	10 (I to X)		Reported in all FMDV-infected areas around the world.
Asia 1	2	Europe-South America (Euro-SA) [12].	-	22	
C	3	Europe–South America (Euro–SA) and Asia [12].	-		Last observed in Kenya in 2004 [15]. It is an uncommon serotype and is not a serotype of concern.
SAT1	13	I (North West Zimbabwe, NWZ), II (South East Zimbabwe, SEZ), III (Western Zimbawe, WZ), IV (East Africa, EA1), V, VI, VII (East Africa EA2), VIII (East Africa EA3), IX, X, XI, XII, and XIII [10,12,16]			Higher nucleotide and amino acid sequence diversity within each other than in the serotype. Intratypic variation is more common in SAT types than in European serotype O.
SAT2	14	I in South Africa, I and II in Zimbabwe, III in Botswana, IV and IX in Kenya, V in Ghana and Nigeria, VI in Gambia, VII in Central African Republic, VII in Saudi Arabia, VIII and X in Zaire, IV, XII, XIII, and XIV in Ethiopia, XI in Angola, XIII in Rwanda, X and XII in Uganda, XIII in Sudan. [10,12,16]			Apart from African territories, it is observed in countries south of the Sahara desert, and in the Northern African and Middle East region, such as Libya, Egypt, the Palestinian Autonomous Territories (PAT), and Bahrain [17].
SAT3	5	I (South East Zimbabwe, SEZ), II (Western Zimbabwe, WZ), III (North West Zimbabwe, NW), IV and V (East Africa, EA) [10,12,16]	25	-	SAT3 has relatively less epidemiological coverage on the continent and rarely affects buffaloes. It is found in Uganda and Zimbabwe.

**Table 2 viruses-15-00797-t002:** Different types and generations of FMDV vaccines and their evolutionary period.

Generations of FMDV Vaccines	Vaccine Types	Evolutionary Period	Major Scientific Techniques and Activities Undertaken	Reference
1st generation	Inactivated vaccines	1930–1940	Inactivation method—formaldehyde antigen; using slice of virus-infected cattle tongue.	[25]
		1950	Inactivation method—formaldehyde antigen; in vitro culture of FMD virus in bovine tongue epithelium (for large-scale production).	[26]
		1970s	Antigen—FMD virus grown in monolayers of BHK cells; cell inactivation—binary ethyleneimine (BEI) and use of oil adjuvant; use of BHK cells for high virus yield and low cell density.	[24]
		~2000	Inactivation method—formaldehyde, binary ethyleneimine (BEI), N acetylenimine, non-chemical methods (endonucleases and hydrostatic pressure). Antigen—attenuated FMD virus by de-optimization or gene deletion.	[32,33]
	Live attenuated vaccines	1960s	Attenuation methods—conventional—cell culture—BHK 21 advantage—higher immunogenicity. Disadvantage—risk of reversion, thermo liable, limited duration, higher cost of production, DIVA.	[26,34,35]
		~2000	Attenuation method—novel method—de-optimization or gene deletion (deletion of full L-pro, deletion of SAP from L-pro, using closely related L-pro of other viruses). Advantage—more stable, less risk of reversion, high neutralizing antibody titers, NS proteins are potent T cell epitopes.Disadvantage—needs high-tech technology for preparation.	[36,37,38]
	Advantage—higher immunogenicity (confer humeral and cellular immunity). Shortcomings of the classical inactivation—short-lived immunity; formulated vaccines need adequate cold chain; risk of recombination with the wild strains; difficulties growing certain serotypes and subtypes well in cell culture for vaccine production; high cost; reversion of the pathogenicity; and DIVA.			[5,39]
2nd generationgenetic engineering vaccine	Genetically engineered subunit vaccines	Advanced after 2000	Method—using recombinant DNA technology and reverse genetics. Antigens—single linear or complex peptides, mostly structural and non-structural proteins (encoded to B and T cell epitopes). Multiple epitopes, 3D conformation epitopes, and utilization of dendritic cells. Expression of target proteins or peptides via bacteria, baculoviruses, mammalian cells, and transgenic plants.	[40,41,42]
	Synthetic peptide vaccine [43]		A synthetic polypeptide designed to resemble a natural epitope (synthetic peptide vaccine for FMD [44].	
	Advantage: relatively low-cost production, stability, and producibility on a large scale. Shortcoming: dependent on carrier proteins.			
	Recombinant vaccine	Advanced after 2000	Method—recombining FMDV immunogenic viral structural proteins with other viral vectors (chimeric vaccine) Vectors; porcine or bovine parvovirus, canine or human adenovirus, herpes virus, fowl pox phages [45].	[46,47,48]
3rd generation	DNA vaccine	This research type peaked after 2015	Method—plasmid with promoter for the target gene (a gene of interest) expression.	
	Virus-like particles (VLPs)	More research of this kind encountered after 2015	Method—transfer of the sequence of the FMDV capsid into a replication-defective human adenovirus type 5, baculovirus, plants, yeasts, other multiple viruses (chimera), and the recombinant expression via eukaryotic and prokaryotic cells.	[45,49,50,51,52,53,54]
	Advantages: stimulate both T and B cells; do not hassle the immune system of the vaccinated animal; safe to use; easy to manufacture and produce; stable and do not require a cold-chain facility; can include marker genes with DIVA capability; can be modified quickly to include field strain sequences; and can contain multiple antigenic sites. Shortcomings: lower immunogenicity and requires advanced biotechnological platforms.			[55,56]

**Table 3 viruses-15-00797-t003:** Experimentally known neutralizing epitope regions of selected serotypes of FMDV.

Capsid Proteins	Serotypes				Structural Function
	O	C	A	Asia 1	
**VP1**	133–157 [114] 200–213 [115] 40–60 [116]	138–150 [117], 195–206, 43–48 and 170 [118]	140–160, 169 (A10) [119] 198 (A5) [120]	142 [121]	cell epitopes
**VP2**	70–78 [114] 131–134 [116]	70–80 [118]	72(A5), 79(A5) [120]	67–79 [121]	
**VP3**	56–58 [116]	58–61 [118]		58, 59, 218 [121]	

**Table 4 viruses-15-00797-t004:** Methods used for prediction of epitope regions in FMDV serotype SAT2.

FMDV Serotype	Epitope Prediction Method	Description	References
SAT2	Chicken single-chain antibody fragments	A single-chain variable fragment (scFv) phage display library on the chicken immunoglobulin genes applied to map neutralizing and putative epitopes in FMD SAT2. For this, three unique soluble binders to the SAT2 virion were selected from the Nkuku^®^ chicken scFv phage-displayed library. The result indicated that only scFv2 was capable of neutralizing the ZIM/7/83 virus and the two others for putative binding sites to the virus.	[136]
	Monoclonal-antibody (MAb)-resistant mutants	This method was used for mapping antigenic sites on FMDV, and topotypically different strains of SAT2 FMDV used for identification of unique antibody-binding footprints on the capsid. The result shows antigenic epitope residues 71 to 72 of VP2 and other multiple epitopic sites on the capsid VP1 of an SAT2 FMDV.	[137]
	In silico prediction	Carried out using freely accessible, web-based B cell epitope prediction servers. Efficiency and accuracy of these bioinformatic programs werwe evaluated in experimentally known epitopes of FMDV. Michelle et al. reported different novel epitopes on the SAT2 3D capsid structure of FMDV using in silico.	[125,136]

**Table 5 viruses-15-00797-t005:** Experimentally known epitope regions on FMDV serotype SAT2.

Epitope Types	Structural Proteins and Epitope Sites		
	VP1	VP2	VP3
B cell	48–50 [137] 140–150 [62] 147–149 [142] 156, 158, 159 [136]	71–72, 133–134 [137] 89–105 [139]	55–88, 176–186, 208 [62]
T cell	21–40, 161-C terminal [143] 135–144, 150–160 [8] 210 [137]	49–68, 113–132, 179–198 [62]	130–134 [8]

**Table 6 viruses-15-00797-t006:** Common immune adjuvants and their principles for foot-and-mouth disease vaccine.

Type of Adjuvant	Main Component	Main Function
Aluminum salt adjuvant [160]	Aluminum hydroxide, aluminum phosphate, and alum	Aluminum adjuvant mainly induces humoral immune response and stimulates TH2 type response.
Oil Emulsion [161]	Complete Freund’s adjuvant and incomplete Freund’s adjuvant et al.	Persistent release of immunogens from oil droplets and stimulation of local immune response.
Poly I:C [162,163]	Poly I:C	Promotes the maturation and differentiation of T cells and DC cells in the body, and enhances the phagocytosis activity of macrophages and cytotoxic effect of NK cells.
CpG-ODA adjuvant [164]	Oligo deoxy-nucletides	Promotes the proliferation and differentiation of B cells, NK cells, dendritic cells, and macrophages, and stimulates TH1 immune response by activating antigen-presenting cells to secrete a variety of cytokines such as IL-6.
Chinese herbal adjuvant	Sugar, glycosides, and other effective ingredients	Chinese medicinal materials such as white fungus: At the same time, it can up-regulate the TH1/TH2 immune response [165]. Propolis adjuvant: By enhancing the role of macrophages, the body, in turn, promotes the immune response to antibodies [166]. Astragalus and other Chinese medicinal materials: Astragalus polysaccharide powder can stimulate the immune function of T and B lymphocytes [167]. Plant saponins (Quil A): Serum cytokine levels and T lymphocyte proliferation rate were significantly increased [168].

## Data Availability

Not applicable.

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
