# Peer review of "B and T Cell Epitopes of the Incursionary Foot-and-Mouth Disease Virus Serotype SAT2 for Vaccine Development"

_viruses, 2023, doi:10.3390/v15030797_

Round 1

Reviewer 1 Report

In the introduction, please describe (Table and figure can be made) three generations of vaccine and their evolution with time.

Please make a table having information about all the serotypes/subtypes of FMDV.

The immune response of vaccines depends upon adjuvants and vaccine delivery along with the antigens/epitopes. Please also discuss about the known and possible adjuvants and vaccine delivery methods that can be used with T-cell and B-cell epitope vaccines.

Please include information (table or figure) about all the known vaccines against FMDV and their limitations.

Figure 4 and its description are not adopted but are almost copied from the previous publication. Please remove this figure and make a new figure with a detailed description.

Author Response

Dear respected reviewer,

We really appreciate the critical points you raised in the review article.  Accordingly, we took much time and due emphasis on your constructive comments and suggestions. So we have revised our manuscript. And, please kindly find the attached point-by-point responses.  

Reviewer 2 Report

I reviewed the manuscript entitled “ B and T Cell Epitopes of the Incursionary Foot-and Mouth Disease Virus Serotype SAT2 and Their  Promising Impact on Vaccine Design and Development”. In this manuscript authors present a perspective about the potential use of epitope based- vaccines for the protection against FMDV SAT2 serotype.

Overall, I think the subject of this review is interesting. However, I also consider that based on the current scientific evidence, it is a research subject on development.

After reviewing the manuscript, I consider that different topics should be included to improve the quality of this manuscript.

A)   I think more information regarding the genetic diversity of this serotype should be added, including the problems with this serotype in some countries like Uganda (Mwiine et al., 2019, doi.org/10.1111/tbed.13249), where different SAT 2 topotypes were  found co-circulating in the same period of time , reflecting the epidemiological complexity of this problem.

B)   Include more information regarding the evolutionary mechanisms of FMDV. How natural selection is acting on B and T epitopes or how recombination may be affecting these sites. Are these sites under strong positive or negative selection?

C)   Include a detailed description of experimental trails involving epitope vaccines in FMDV? Highlighting the ones conducted in relevant species (natural hosts).

Author Response

(The authors gave the same response as above.)

Round 2

Reviewer 1 Report

The authors have addressed all of the comments.

Reviewer 2 Report

I like to thank the authors for their responses. At this point, I don't have more concerns about this study.